# A Semi-Autonomous Hierarchical Control Framework for Prosthetic Hands Inspired by Dual Streams of Human

**DOI:** 10.3390/biomimetics9010062

**Published:** 2024-01-22

**Authors:** Xuanyi Zhou, Jianhua Zhang, Bangchu Yang, Xiaolong Ma, Hao Fu, Shibo Cai, Guanjun Bao

**Affiliations:** 1College of Mechanical Engineering, Zhejiang University of Technology, Hangzhou 310023, China; zhouxuanyi@zjut.edu.cn (X.Z.); mxlslf@126.com (X.M.); fhneversettle@163.com (H.F.); ccc@zjut.edu.cn (S.C.); 2Key Laboratory of Special Purpose Equipment and Advanced Processing Technology, Ministry of Education and Zhejiang Province, Zhejiang University of Technology, Hangzhou 310023, China; 3State Key Laboratory of Chemical Engineering, College of Chemical and Biological Engineering, Zhejiang University, 38 Zheda Road, Hangzhou 310027, China; 4School of Mechanical Engineering, Beijing University of Science and Technology, Beijing 100083, China; jhzhang@ustb.edu.cn

**Keywords:** prosthetic hand, control strategy, grasp, manipulation, human inspired

## Abstract

The routine use of prosthetic hands significantly enhances amputees’ daily lives, yet it often introduces cognitive load and reduces reaction speed. To address this issue, we introduce a wearable semi-autonomous hierarchical control framework tailored for amputees. Drawing inspiration from the visual processing stream in humans, a fully autonomous bionic controller is integrated into the prosthetic hand control system to offload cognitive burden, complemented by a Human-in-the-Loop (HIL) control method. In the ventral-stream phase, the controller integrates multi-modal information from the user’s hand–eye coordination and biological instincts to analyze the user’s movement intention and manipulate primitive switches in the variable domain of view. Transitioning to the dorsal-stream phase, precise force control is attained through the HIL control strategy, combining feedback from the prosthetic hand’s sensors and the user’s electromyographic (EMG) signals. The effectiveness of the proposed interface is demonstrated by the experimental results. Our approach presents a more effective method of interaction between a robotic control system and the human.

## 1. Introduction

It is anticipated that future humanoid robots will perform various complex tasks through communication with human users [1]. Despite significant advancements in prosthetics technology in recent years, only 50 to 60 percent of amputees are willing to wear prosthetics [2], with rejection rates as high as 40 percent [3]. Brain dynamics experiments may provide insight into why this “prosthetic rejection” occurs [4]. Since prosthetic hands are less comfortable than human hands, using them imposes great cognitive burden on amputees, leading to brain fatigue and psychological frustration that ultimately results in prosthetic hand rejection [5].

Recent studies shed new light on the concept of “cognitive load” when using prosthetic hands. Prolonged use of prosthetic hands while handling tools can increase the strength of (electroencephalographic) EEG alpha waves in the brain [6]. This process can lead to an increase in cognitive load, resulting in fatigue and reduced responsiveness to other objects [7], which is the key reason for the high rejection rate of prostheses. The significant increase in EEG alpha-wave power indicates that users consciously exert greater control over their prosthetic hands [8]. This phenomenon is not observed in ordinary individuals when using their hands naturally and skillfully. User experience research reports confirm the existence of “cognitive load” [9]. Equipping prosthetic hands with sensor feedback to reduce the visual dependence of amputees is a promising solution to alleviate cognitive load [10].

An inspiration for reducing the “cognitive load” is the ventral stream and dorsal stream, which are two visual systems hypotheses of the human brain. Recently, evidence was provided for a functional segregation of dorsal and ventral streams, supporting the hypothesis [11]. Anatomical studies have proved the interaction between ventral and dorsal streams, especially for skilled grasping [12]. As the demanding precision of the grasp increases, these physiological interconnections gradually become more active [13,14].

The ventral stream, known as the “what pathway,” is primarily responsible for object recognition and perception. The dorsal stream, referred to as the “where pathway,” is primarily involved in the processing of spatial awareness and movement guidance [15]. The primary function of the dual stream is shown in Table 1. In the context of controlling prosthetic hands, the ventral stream can be utilized to extract information about the user’s hand–eye coordination and their viewing field. By analyzing the information, the controller can determine the intention behind the user’s movements and enable the manipulation of primitive switches in the variable domain of view. The dorsal stream plays a crucial role in integrating visual information with motor control. In the context of controlling prosthetic hands, the dorsal stream enables precise force control. Moreover, the dorsal stream needs to obtain detailed information about the identity of the object stored in the ventral-stream region when object attributes require complex fine-tuning of the grasp. Correspondingly, the ventral stream may obtain the latest grab-relevant information from the dorsal-stream region to improve the internal representation of the object. Based on the hypothesis, incorporating both the ventral- and dorsal-stream principles into the control framework of prosthetic hands makes it possible to create a more sophisticated and intuitive control system. The ventral stream contributes to the perception and recognition of objects, while the dorsal stream facilitates movement guidance and precise force control. This framework should improve the performance of prosthetic hands and user experience.

## 2. Related Work

Prosthetic hands primarily rely on EMG signals for control [16,17]. Although this method effectively utilizes residual muscles in the amputated limb, the absence of corresponding tactile feedback necessitates users to rely on visual compensation, resulting in an increased cognitive load. This phenomenon has emerged as a significant factor in prosthetic rejection among amputees. To address this issue, researchers have developed a gaze-training method known as Gazing Training. This method assists amputees in adapting to prosthetic hand usage by reducing the need for conscious control and alleviating cognitive load [18]. However, it is important to note that while Gazing Training demonstrates partial success in mitigating cognitive load during the rehabilitation process, it does not completely eliminate the underlying challenge [19].

Therefore, some researchers aspire to combine intelligent autonomous control methods with human users’ electromyographic (EMG) signals by integrating control techniques from the field of robotics [20,21]. This integration aims to establish a semi-autonomous controller that combines autonomous control capabilities with human EMG signals. The objective of this semi-autonomous control approach is to partially or even entirely transfer the cognitive load to the controller, enabling autonomous assistance for amputee patients in accomplishing daily tasks. With advancements in both the understanding of the human brain and computer technology in recent years, this is possibility gradually transforming into reality [22].

In 2015, Markovic proposed a framework for a semi-autonomous controller, which consists of an autonomous control unit and an EMG control unit [23]. The autonomous control unit employs computer vision sensors to capture depth and red green blue (RGB) information, as well as proprioceptive feedback from the prosthetic hand, for data processing and information fusion. On the other hand, the EMG control unit utilizes electromyographic signals to reflect user intentions and facilitate manual control of the prosthetic hand. The combination of the EMG control unit and the autonomous control unit forms a semi-autonomous controller that integrates autonomous control capabilities with human EMG signals, allowing for switching between control modes [24].

Bu proposed a visually guided approach to assist patients in achieving semi-autonomous manipulation of prosthetic hands, with the primary goal of alleviating users’ cognitive load [25]. Chunyuan introduced global visual information based on EMG signals [26]. Machine vision was employed to extract object shape, size, type, and appearance information, which was then integrated with pre-hand shape for joint motion planning using visual and dual antagonistic-channel EMG signals. Wang proposed an RNN network by incorporating features of the user’s gaze point [27]. The semi-autonomous control of the prosthetic hand is achieved by the automatic recognition of the used tools and motion primitives. The integration of computer vision into the semi-autonomous control of prosthetic hands imposed significant computation demands on the controller [28]. To address the issue of computational load in prosthetic hand operations, Fukuda proposed a distributed control system to enhance the real-time performance of the semi-autonomous controller [29]. Moreover, the inertial measurement unit (IMU) was employed to obtain the prosthetic hand’s state, while the visual system was utilized to perceive object states [30]. Vorobev proposed a semi-autonomous control method for prosthetic hands. Motion commands were transmitted to the prosthetic hand’s main controller using sensors triggered by foot movement inside the shoe [31].

In summary, addressing the issue of cognitive load, the Gazing Training approach is proposed as a rehabilitation training method for amputees to adapt to prosthetic hands [32]. The Gazing Training partially mitigates the cognitive load but does not completely eliminate the problem. In recent years, some researchers have proposed a semi-autonomous control strategy from the perspective of robotics engineering. In these approaches, the controller acquires information about object shape, size, etc., and translates it into corresponding motion primitives, while the user’s EMG signals generate grasping commands for the prosthetic hand. This semi-autonomous hierarchical control strategy aims to transfer a portion of the user’s cognitive load to the controller, thereby reducing the user’s alpha-wave power. Building upon the foundation of traditional semi-autonomous controllers, this paper primarily presents two optimizations to the semi-autonomous hierarchical control strategy for prosthetic hands. The innovations of the paper are summarized as follows:

(1)A controller is constructed based on the pathway of the human ventral–dorsal nerves. Object semantic segmentation and convolutional neural network (CNN) recognition are categorized as the ventral stream, while the motion tracking of the limb is introduced as the control of the dorsal stream. Moreover, the dorsal stream and ventral stream are integrated to ensure accurate motion primitives.(2)In order to reduce the cognitive burden, a semi-autonomous controller is proposed. Feedback of the prosthetic hand is integrated to enhance the perceived experience. EMG signals of the user are obtained to realize the human in the loop control.

## 3. Methodology

In the guidance stage of the human visual neural stream, the perception-motion guidance is carried out by the ventral stream to recognize and locate objects. The ventral-stream information is matched to the related object memory and long-term action primitives. Inspired by the ventral stream, the prosthetic hand control system is applied to locate the object. Moreover, the initial user intentions are obtained through the posture of the grasping task. The visual information of objects can be further obtained through the CMOS image sensor of the head-mounted device. Since the CMOS image sensor in the headset follows the vision of the human user, the headset is able to locate the objects and prosthetic hands.

In the human dorsal-stream guidance stage, the human user guides the prosthetic hand to grasp and manipulate in a vision-aided manner. At the same time, the CMOS image sensor of the headset will obtain the real-time distance between the prosthetic hand and the object. When the prosthetic hand approaches the object, the controller will drive the prosthetic hand to perform grasp motions with the EMG signal of the human body simultaneously. At this stage, humans mainly use the information of the dorsal neural stream for guidance. Based on the characteristics of dorsal-stream information, the position and force cloud of the grasping process are fed back to the human user. Finally, the EMG signals of the human are combined to realize the human in the loop control. The framework of two nerve streams of visual information is shown in Figure 1.

### 3.1. Task-Centric Planning

The task planning is divided into two parts: motion primitive and sequence planning. The controller is proposed in the task-centric task, which is inspired by the two visual streams, realizing the sets and independent planning control of prosthetic hand motion primitives. Firstly, multiple motion primitives are stored in the semi-autonomous controller. Different motion primitives are applied in the semi-autonomous controller for the grasping features of different objects. The object’s category and the prosthetic hand’s configuration are input into the controller as decision factors. The controller drives the prosthetic hand to complete the grasping task. In the decision-making stage, it is necessary to analyze the object character and obtain the location information in the scene. The decision is calculated by the convolutional neural network and semantic analysis. In order to realize real-time control, the SSD-Mobilenet-V2 convolutional neural network is used to realize semantic segmentation and object recognition for visual images. SSD-Mobilenet-V2 is used for semantic segmentation while Mobilenet is used for object recognition. The Single-Shot Multi-box Detector (SSD) neural network generates constraint squares of fixed size through the forward-propagation CNN network, compared to the intersection over union (IOU) of different Anchor boxes generated on the feature graph, obtaining boundary boxes close to 0.5.

Images captured from CMOS image sensors are segmented and associated with semantics. Specifically, after obtaining the boundary boxes through the SSD method mentioned above, a mobilenet-V2 convolution network is introduced to identify objects in the boundary boxes and generate index information. To improve the operational efficiency of the neural network for embedded devices, the mobilenet-V2 convolutional network is adopted to simplify the calculation process through a deeply separable convolutional operation method. The main advantage of this method is that Linear Bottleneck replaces ReLU to activate a function, achieving channel reductions depth wise and dimension reductions point wise, respectively. At the same time, to reduce the effect of feature reduction caused by linear bottleneck dimension reduction, the addition and nonlinearity of features are realized by inverted residuals. Similarly, an inverted residual neural network has the property of “short-circuit”. The procedure has been simplified as the lightweight software requirement of mobile devices.

In different scenes, objects will correspond to a different set of motion primitives. Taking three grasping and manipulating tasks as examples, the varieties of manipulated primitives involved in this paper are shown in Table 2.

For different grasp and manipulate tasks, motion primitives are required for the different manipulate time sequence planning. Gesture 1 is no contact and no motion of the hand. Gesture 2 is motion of the hand without and within the hand. Gesture 12 is the within-hand movement [33]. To ensure the suitable sequence of motion primitive switching, sequential planning uses the end state of the previous primitive as the beginning state for the next primitive. Considering a common motion in daily life, “object grasping”, the sequence of manipulation can be planned as follows:(1)The beginning of the task. The initial motion primitive is in the free state, which is gesture 1.(2)Object recognition stage. When the target recognition is completed, the prosthetic hand forms the pre-grasp posture according to the feature information of the object. If the target image is lost, it is estimated that the user gives up grasping, and the prosthetic hand restores to the free primitive state.(3)Pre-grasp stage. The head-mounted CMOS image sensor obtains the spatial position relationship between the prosthetic hand and the object in real time. When the space distance between the hand and object is less than the threshold value, it is judged that the hand and object are in contact and enter the grasping primitive stage. When the space distance between the hand and object is greater than the threshold value, the hand and object are considered to be separated and return to the pre-hand type stage. The user can change his view field to restore the original free primitive at this stage.(4)Grasp stage. Control is performed on the prosthetic hand after fusing EMG signal and object vision information feedback.(5)After grasping, the prosthetic hand ends the grasping task and returns to the initial stage. After completing the grasping task, the prosthetic hand is controlled to separate from the object and restored to the initial free primitive state, changing the view field of the CMOS sensor. In case of unsuccessful separation of the hand and object or an emergency, voice command can perform an emergency reset. We restrict the prosthetic hand from switching directly to the free primitive state when performing grasp primitives for the user’s safety. The aforementioned manipulate sequence planning for the controller is shown in Figure 2.

### 3.2. Precise Force Control Strategy

The precise force control of the prosthetic hand during contact is proposed for object grasping. In the grasping process, human users are able to participate in the control loop, forming the human-in-loop control mode. The pressure cloud image is shown in the headset. With the pressure feedback, the human users can adjust the grasping force to realize closed-loop control between the human and the prosthetic hand.

Human signals are generated by the flexor digitorum profundus. Signal preprocessing is carried out, including signal amplification, peak-to-peak detection, envelope processing, mean filtering, Fourier transform, and bandpass filtering. The value of the EMG signal is obtained in real time.

The STM32 microcontroller is applied as the central semi-autonomous controller. An Arduino microcontroller was used to obtain EMG values of flexor digitorum Profundo in real time. These two microcontrollers communicate through a serial port and transmit data in hexadecimal format. The central semi-autonomous controller will request for the intensity of the EMG signals obtained by the Arduino controller in real time. Considering the diversity of tasks, environments, and personal behaviors in grasping, the semi-autonomous controller will continuously detect the action potential of the user’s digital-flexor deep muscle. A typical original EMG signal and bandpass filtering EMG intensity are shown in Figure 3. The proportional control output u will be obtained when the EMG intensity P exceeds the threshold, where $k_{1}$ is the proportional coefficient.
(1)$$u=k_{1}P$$

The control output value u of the semi-autonomous controller is transmitted to the prosthetic hand system through Bluetooth. The prosthetic hand system converts the proportional control output u into pulse-width modulation (PWM) signal. It controls five servo motors to realize the movement of the finger joint of the prosthetic hand. Pressure sensors are embedded in each finger to read the contact state between the prosthetic hand and the object. The haptic pressure distribution cloud is transmitted to a semi-autonomous controller via Bluetooth built into the prosthetic hand system. To tackle the challenge of providing grip force feedback in prosthetic hands, this study leverages computer graphics techniques utilizing the open-source graphics library OpenGL. Within the semi-autonomous controller, a pressure cloud map of the prosthetic hand is generated. This map is subsequently projected onto the user’s retina through a head-mounted device, facilitating closed-loop control. The fundamental approach involves the transmission of pressure sensor information from the prosthetic hand system to the semi-autonomous controller via the embedded Bluetooth of the microcontroller unit. The semi-autonomous controller, in turn, employs OpenGL to render the pressure cloud map. The combined controller information and prosthetic hand data are then projected onto the user’s retina using the micro display of the head-mounted device.

Users will dynamically adjust the incremental control output of the prosthetic hand system based on real-time feedback from the controller and visual perception. According to Equation (1), once the proportional control output *u* is generated, the semi-autonomous controller will continuously monitor the user’s electromyographic signals. If the user perceives that a stable grasp has not yet been achieved, the proportional control output *u* will persistently accumulate control output increments.
(2)$$u_{i}=u_{i-1}+k_{2}P$$
where $k_{2}$ is the incremental coefficient, which plays a role in amplifying and shrinking the signal during the incremental proportional control. The procedures of the proportional control mode are shown in Figure 4.

To solve the grasping force acquisition of prosthetic hands, we draw a pressure cloud map of prosthetic hands in the semi-autonomous controller, using computer graphics technology (OpenGL). This process costs less in terms of CPU resources. The pressure cloud figure is displayed in a headset and projected onto the user’s retina. In this way, the human brain is connected to the control loop of the prosthetic hand. Rendering the pressure cloud image in the semi-autonomous controller mainly uses the GLWight library under the Qt framework; the generated rendering image is transmitted to the head-mounted device through an HDMI cable. The micro-display device in the headset uses a Vufine+ wearable display for visual projection.

OpenGL’s geometric shader function interface is a four-channel array representing R (red region), G (green region), B (blue region), and α (transparency component). The following formula will convert the contact pressure information transmitted into the OpenGL into a cloud map. The pressure cloud figure will be displayed on the head mount display. When the contact pressure value $c_{i}$ is less than 0.5-times the contact pressure threshold L, the data relationship of contact pressure information to the cloud map is shown in Formula (3):(3)$$\left\{\begin{array}{c} R_{i}=0\\ G_{i}=\frac{255c_{i}}{0.5L}\\ B_{i}=255-\frac{255c_{i}}{0.5L} \end{array}\right.$$

When the contact pressure value $c_{i}$ is larger than 0.5-times of the contact pressure threshold L, the data relationship of contact pressure information to the cloud map is shown in Formula (4):(4)$$\left\{\begin{array}{c} R_{i}=\frac{255c_{i}}{0.5L}\\ G_{i}=255-\frac{255c_{i}}{0.5L}\\ B_{i}=0 \end{array}\right.$$
where $c_{i}$ is the contact force value of the ith finger; L is the contact force threshold; $R_{i}$, $G_{i}$, $B_{i}$ represent the color components of the ith finger in red, green, and blue cloud images, respectively.

Tailored to specific grasping or operational tasks, the semi-autonomous controller dynamically monitors the real-time electromyographic signal strength of the user’s deep flexor muscles. It triggers the electromyographic control phase when this intensity surpasses a predefined threshold. Considering that deep flexor muscle signals originate from the deeper muscle groups of the human body and surface electrode signal acquisition is susceptible to interference, various techniques, including differential amplification, peak detection, envelope processing, mean filtering, Fourier transformation, and bandpass filtering, are employed to extract real-time electromyographic signal intensity values. These intensity values are communicated in real time via serial communication to the STM32 embedded in the semi-autonomous controller for proportional control.

When the prosthetic hand contacts the target object, the contact force value and force distribution position of the tactile sensor embedded in the prosthetic hand are changed. This tactile information will be processed by the processor embedded in the prosthetic hand and then transmitted to the semi-autonomous controller, which uses OpenGL to draw the pressure cloud. A micro-display on the headset projects the pressure cloud image onto the user’s retina, enabling the user to know the motion state and contact force distribution of the prosthetic hand system in real time for further incremental control until the user confirms the stable grasp, as shown in Figure 5.

## 4. Experiment

This section sets up a grasping platform for prosthetic hands to verify the feasibility of using the semi-autonomous controller and its robust performance. Users can realize grasping and manipulating tasks of a prosthetic hand using the semi-autonomous controller in the way of “hierarchical visual stream driven”.

As shown in Figure 6 (from the recorder’s perspective), the subject of this experiment is a 25-year-old Chinese adult male with normal vision. Elements of the semi-autonomous control framework are listed below:Jetson Nano (Nvidia, Santa Clara, CA, USA), which has 128-core NVIDIA Maxwell™ architecture GPU and Quad-core ARM^®^ Cortex^®^-A57 MPCore processor and the semi-autonomous controller are integrated in the Jetson Nano.WX151HD CMOS image sensor (S-YUE, Shenzhen, China), which has 150-degree wide angle.ZJUT prosthetic hand (developed by Zhejiang University of Technology, Hangzhou, China) is equipped with 5 actuators.DYHW110 micro-scale pressure sensor (Dayshensor, Bengbu, China) is integrated in the prosthetic hand to obtain the touch force. It has a range of 5 kg, and the combined error is 0.3% of the full scale (F.S.).Vufine+ wearable display (Vufine, Sunnyvale, CA, USA) is a high-definition, wearable display that seamlessly integrates with the proposed control framework.

On the right side of the experimental setup is the semi-autonomous controller independently developed by the laboratory for this study. This controller follows the user’s multi-modal information and facilitates the grasping function of the prosthetic hand. The detailed structure of the semi-autonomous controller was thoroughly discussed in Section 3. Additionally, the prosthetic hand system, also independently developed by the laboratory, is affixed to the damaged hand of the mannequin wearing the semi-autonomous controller. Other components within the experimental environment include a Graphic User Interface (GUI) on a personal computer. This GUI serves the convenience of subjects, experiment operators, safety officers, and recorders, enabling them to monitor the state of the semi-autonomous controller and make timely adjustments. The object being manipulated in the experiment is a plastic beverage bottle, mimicking the user’s routine grasping actions for daily beverage needs. The experiment recorder controls and adjusts the process based on the recorded experimental footage, as illustrated in Figure 7.

### 4.1. Prosthetic Hand Grasping Experiment

In their daily lives, amputees often grasped objects with their prosthetic hands. The purpose of the experiment is that the semi-autonomous controller can assist the user in grabbing the “bottle” naturally under the multi-mode interaction between the semi-autonomous controller and the human user. The grasping process is as follows, which can be seen in Figure 8:(1)The semi-autonomous controller was worn by a dummy model. A prosthetic hand, electromyographic electrode, and head-mounted device on the human subject are used to obtain the human EMG signal and project the signal to the human eye. (Figure 8.1).(2)The human subject attached the prosthetic hand to his left hand and exhaled, “grab the bottle”. The subject was looking at the bottle with his arm close to it. The head-mounted device will perform visual semantic segmentation and convolutional neural network recognition for the “bottle” in this process. According to the information returned by the CMOS image sensor and the speech task library, the prosthetic hand can switch the combination of motion primitives. (Figure 8.2).(3)When the human subject’s arm comes close to the “bottle”, it generates an EMG signal. The semi-autonomous controller implements proportional and incremental control under a specific motion primitive, depending on the EMG strength, until the user is instructed to “determine” the prosthetic hand motion. During this period, users can observe the changes in the pressure cloud of the prosthetic hand in real time. (Figure 8.3–6).(4)The experimental subjects grasp the “bottle” and place it in another position on the table. After placement, the prosthetic hand is released and reset by voice. (Figure 8.7–8).

### 4.2. Prosthetic Hand and Human Hand Coordinative Manipulation Experiments

This experiment aims to verify that the semi-autonomous controller can assist humans in completing the manipulation of “screw the bottle cap” naturally under the multi-mode interaction between the semi-autonomous controller and the human user. The cooperative manipulation experiment of the prosthetic hand and human hand can be seen in Figure 9:(1)The semi-autonomous controller was worn by the dummy. A prosthetic hand, EMG electrode, and head-mounted device were attached to the human subjects. (Figure 9.1).(2)The experimental subject fixed the prosthetic hand on his left hand and exclaimed the command “screw the bottle cap” by voice. While the subject is looking at the “bottle”, the user is grabbing the “bottle” with his right hand, and the prosthetic hand with his left arm is approaching the “bottle cap”. The head-mounted device will perform visual semantic segmentation and convolutional neural network recognition for the “bottle” in this process. According to the information returned by the CMOS image sensor and the speech task library, the prosthetic hand can switch the combination of motion primitives. (Figure 9.2–3).(3)When the distance between the prosthetic hand and the “bottle cap” is less than the threshold value, the prosthetic hand will grab the “bottle cap”, and the user drives the prosthetic hand to rotate the “bottle cap” to the set angle through his left arm. (Figure 9.4–7).(4)When the “bottle cap” is not unscrewed and the distance between the prosthetic hand and the “bottle cap” exceeds a certain distance, the prosthetic hand will return to the pre-hand type.(5)Repeat Step 3 and Step 4 until the cap is unscrewed.(6)Place the unscrewed bottle cap on the desktop and command the voice to reset the prosthetic hand. (Figure 9.8–10).

### 4.3. Results

According to the results of two experiments, the proposed two visual-stream-driven manipulation strategy is in line with the natural manipulation rules of human beings and can effectively assist patients in completing familiar grasping and manipulation tasks in daily life. The experimental results are consistent with the expectations.

The experimental results shown in Table 3 combine the user experience and controller characteristics. The grasping task of a prosthetic hand and the human manipulation task described in this paper are similar in the structure form of the control method. In terms of the task layer, voice interaction is combined with the visual neural network. Due to differences in user intention and environment, the processing content will change the output result. At the planning level, the action primitive of the grasping task mainly switches between grasping hand type and pre-grasping hand type; the state transition condition is primarily used to trigger the user’s man-in-loop control. The motion primitives of manipulation tasks switch between multiple manipulators, and the state transition condition is mainly used to trigger the arrangement of action primitives. In the motion control layer, the size of manipulated objects is usually tiny, so the switch of motion primitive is mainly realized through arm movement guidance. There are differences in the size, mass, and type of the objects in the grasping task, which requires higher grasping stability; thus, the force control strategy of the human-in-loop is introduced.

## 5. Conclusions and Future Works

### 5.1. Conclusions

Different from the traditional semi-autonomous controller design, this work is inspired by the two visual streams of the humanoid control strategy. The summary of this framework can be seen in Figure 10. The main contributions are as follows:(1)In terms of the control layer of the motion primitive planning, the traditional object semantic segmentation and CNN recognition are classified as ventral flow, residual arm motion tracking is introduced as dorsal flow, inspired by the human brain. We optimized the information collection method of the motion primitive planning control layer and the state transfer strategy among the movement primitives according to the multimodal information in different stages of ventral flow and dorsal flow.(2)In terms of the force control layer, the issue of the user’s “cognitive burden” is reduced in the existing semi-autonomous control strategy, so this paper takes the user as a high-dimensional controller and the EMG strength of flexor digitorum profundus as the control quantity, giving feedback to the prosthetic hand body state, realizing precise force control with human-in-loop.

### 5.2. Future Works

The proposed control framework, inspired by the human ventral–dorsal stream visual process, currently emphasizes functional grasping actions within the spectrum of human hand operations. However, this biological process represents just one aspect of the myriad ways humans execute gripping tasks, emphasizing functional manipulations. Ongoing neuroscientific efforts delve into understanding how different brain regions guide grasping operations through visual cues. Building upon the current ventral–dorsal stream, future work will explore bio-inspired control strategies, integrating the latest neuroscientific findings related to the user’s dual visual neural pathways. The aim is to deepen the algorithmic sophistication and broaden the spectrum of multimodal information, fostering a more profound integration between humans and machines.

In future investigations, a pivotal area for exploration revolves around the usability and social acceptance of prosthetic hands in various environments, especially social situations. Current technology, while advancing rapidly, may pose challenges in social integration due to its external and conspicuous nature. To enhance user experience and societal acceptance, future work could focus on developing discreet, aesthetically pleasing designs that seamlessly integrate prosthetic hands into social settings. This involves not only technical advancements but also a nuanced understanding of user preferences, comfort levels, and societal perceptions. Exploring materials, form factors, and user-centric design principles could contribute significantly to reducing the stigma associated with prosthetic devices, fostering a more inclusive and socially integrated environment for users. This research direction aligns with the broader goal of not only improving the functional aspects of prosthetic hands but also enhancing the overall quality of life and social experiences for individuals using this technology.

## Figures and Tables

**Figure 1 biomimetics-09-00062-f001:**
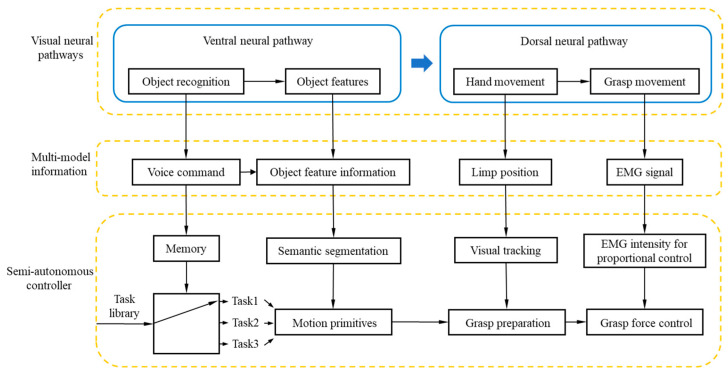
The framework of two nerve streams of visual information.

**Figure 2 biomimetics-09-00062-f002:**
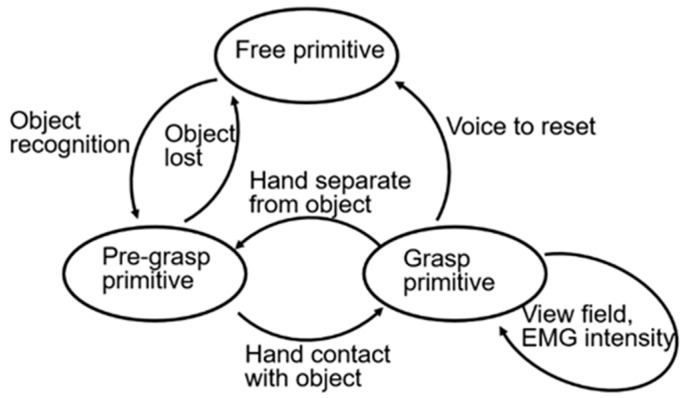
Manipulate sequence planning for the controller.

**Figure 3 biomimetics-09-00062-f003:**
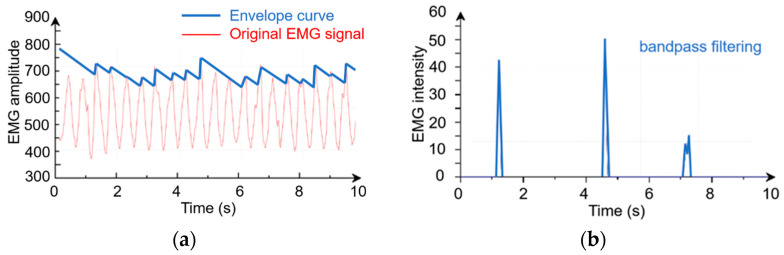
The myoelectric signal. (**a**) Original EMG signal. (**b**) Bandpass filtering.

**Figure 4 biomimetics-09-00062-f004:**
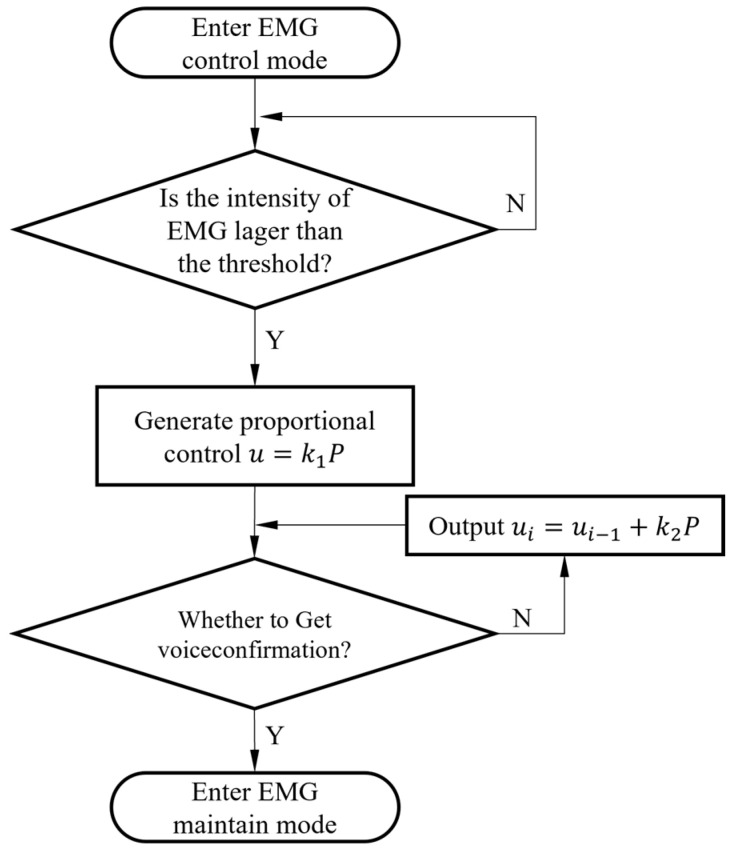
The procedures of the proportional control mode.

**Figure 5 biomimetics-09-00062-f005:**
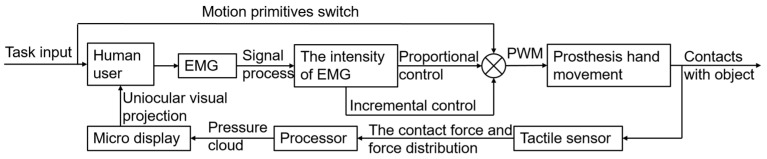
Block diagram for grasping force control of prosthetic hand with the human in the control loop.

**Figure 6 biomimetics-09-00062-f006:**
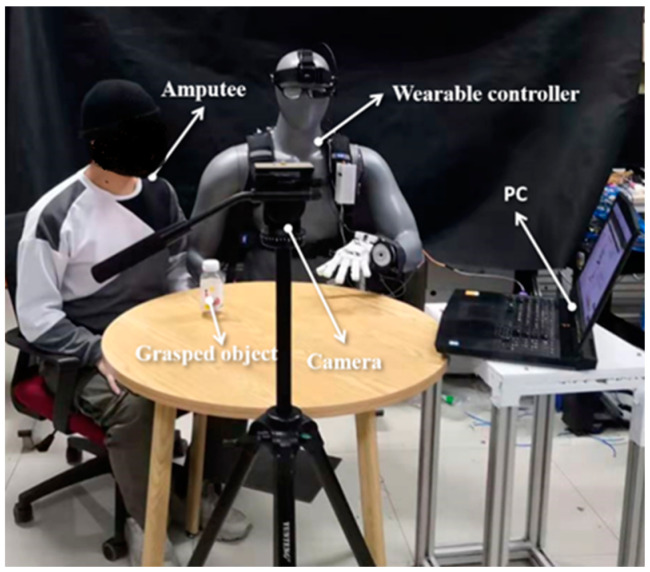
Experimental setup.

**Figure 7 biomimetics-09-00062-f007:**
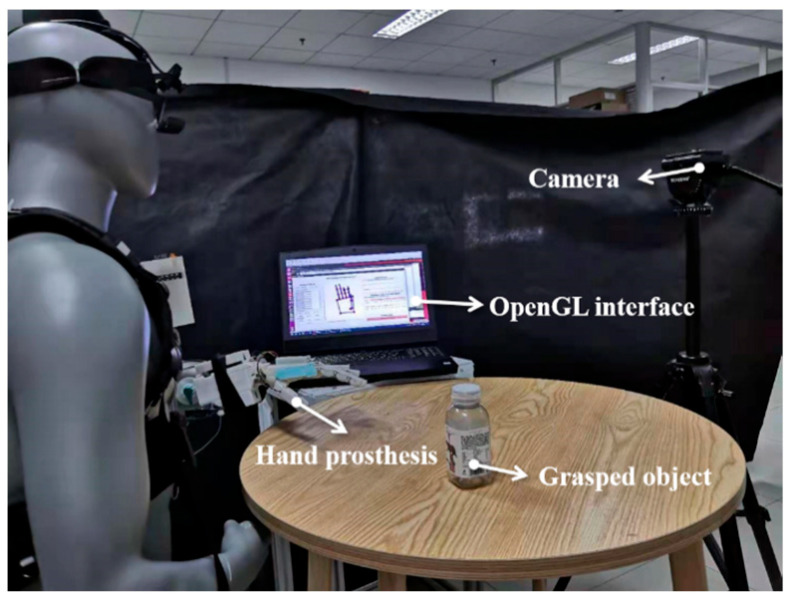
The view field from the subject’s perspective.

**Figure 8 biomimetics-09-00062-f008:**
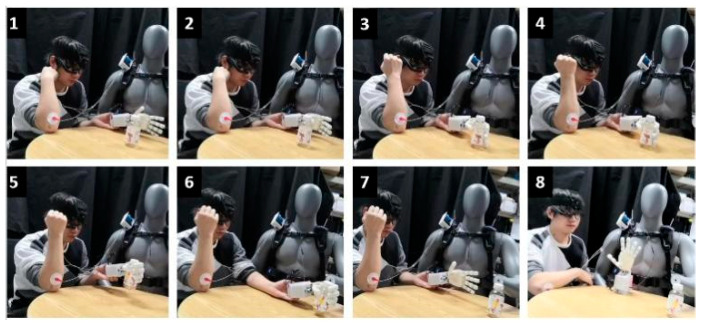
Arm and hand collaborative control experiment.

**Figure 9 biomimetics-09-00062-f009:**
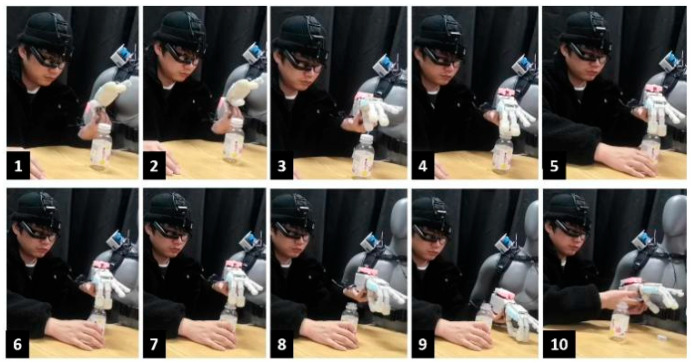
The cooperative manipulation experiment of a prosthetic hand and human hand.

**Figure 10 biomimetics-09-00062-f010:**
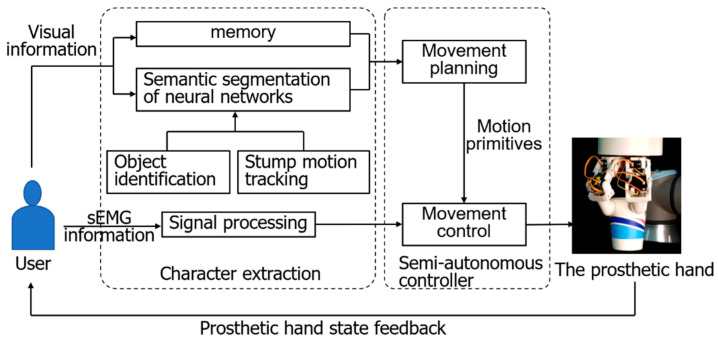
Framework of the semi-automatic controller for the prostheses hand.

**Table 1 biomimetics-09-00062-t001:** The primary function of the dual stream.

	The Ventral Stream	The Dorsal Stream
Function	Identification	Visually guided movement
Sensitive features	High sensitivity to spatial	High sensitivity to time
Memory features	Long-term memory	Short-term memory
Reaction speed	Slow	Quick
Comprehension	Very fast	Very slow
Reference frame	Object-Centric	Human-Centric
Visual input	Fovea or parafovea	Entire retina

**Table 2 biomimetics-09-00062-t002:** The sets of motion primitives.

	Toggle Switch	Screw Cap	Grasp Cap
The rest gesture	gesture 1	gesture 1	gesture 1
Pre-shape gesture	gesture 12	Prepare and pre-envelope	Prepare and pre-envelope
Manipulate gesture	gesture 12	gesture 2	gesture 2

**Table 3 biomimetics-09-00062-t003:** The experimental results.

	Task Type	Task Characteristics
Coordinative movement of hand and arm	Grasp	Force control
Two hands coordination	manipulation	Movements switch

## Data Availability

Data are contained within the article.

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
