# Peer review of "A Semi-Autonomous Hierarchical Control Framework for Prosthetic Hands Inspired by Dual Streams of Human"

_biomimetics, 2024, doi:10.3390/biomimetics9010062_

Round 1

Reviewer 1 Report (Previous Reviewer 1)

Comments and Suggestions for Authors

The paper presents a control strategy for prosthetic hands that aims at reducing the cognitive burden of the user by integrating an EMG-based prosthesis with a vision system to recognize the object to grasp and to call the appropriate motion primitive. Vocal commands are used in various stages of the process. Moreover, the system can give a real-time feedback about the exerted force.

Many problems have been found in the initial version of this paper. The revised version has corrected most of them.

The low originality of the approach is still present. The dual steam idea is not making a real difference. Anyhow, the authors have better explained both the principle and the implementation, so the results are better understood.

They have also indicated that the real application of the method in prosthesis is far because cameras and other needed equipment are not available now.

The user acceptance of the method has not been approached; the authors recognize it should be done in future. In conclusion, it has been recognize the exploratory study level of the proposed system.

Comments on the Quality of English Language

Minor editing necessary

Author Response

The authors would like to express their sincere appreciation to the reviewer for his/her constructive comments and suggestions, and his/her time and efforts spent in helping us to improve the quality and presentation of the paper.

  • Comment:Many problems have been found in the initial version of this paper. The revised version has corrected most of them.

Response: Thank you for your valuable feedback. We hope that the modifications made align with your expectations and address the concerns raised during the review process. We are grateful for your insightful comments, which have undoubtedly contributed to the refinement of our work.

  • Comment:The low originality of the approach is still present. The dual steam idea is not making a real difference. Anyhow, the authors have better explained both the principle and the implementation, so the results are better understood.

Response: We appreciate the helpful feedback provided by the reviewer. We acknowledge the concern regarding the perceived low originality of the dual-stream approach. While we understand this perspective, we would like to emphasize that the dual-stream concept was introduced to enhance the adaptability and versatility of the prosthetic hand control system. We have made efforts to better articulate both the underlying principles and the practical implementation to improve the clarity and understanding of our work.

  • Comment:They have also indicated that the real application of the method in prosthesis is far because cameras and other needed equipment are not available now.

Response: Thank you for your valuable suggestion. We agree that the practical implementation of the method on prostheses requires consideration of the current technological landscape, including the availability of suitable cameras and related equipment. As technology advances, we anticipate that the necessary hardware for our proposed method will become more accessible and refined.

In future work, we plan to keep a close eye on technological developments and collaborate with industry partners to adapt our method to the evolving landscape of prosthesis technology. This may involve exploring alternative sensor technologies or adapting our approach based on the availability of suitable equipment.

  • Comment:The user acceptance of the method has not been approached; the authors recognize it should be done in future. In conclusion, it has been recognize the exploratory study level of the proposed system.

Response: We appreciate the insightful feedback provided by the reviewer. We acknowledge the current limitations in our study regarding this aspect. As recognized by the reviewer, we concur that the user acceptance level of the proposed method requires thorough investigation, which we plan to undertake in future research. Understanding how users interact with and embrace the developed system is vital for its practical applicability and societal integration.

In conclusion, we acknowledge that our study is positioned as an exploratory endeavor, laying the groundwork for further investigations, including in-depth user acceptance studies. We appreciate the constructive critique, and your feedback will significantly contribute to refining our research and ensuring its relevance and impact in practical settings.

Reviewer 2 Report (New Reviewer)

Comments and Suggestions for Authors

This article is modern and relevant. In the modern world, it is very necessary to use bionic prosthetic hands. The experience of warfare in Ukraine and in the Gaza Strip has confirmed the high percentage of amputations during surgical operations in the field. Therefore, the creation of new bionic prostheses for human limbs, especially hands and arms, is a very important and urgent task. The use of visual information to reduce the cognitive load on the prosthesis is a very important factor in rehabilitation treatment after the amputation of a hand. All the above confirms the high relevance of the results obtained in this article.

The article is well-written in good, competent, and understandable English. The structure of the article is traditional. All sections are written in sufficient volume. The research methods are described well enough, and the results are clearly presented. In the Introduction section, there is enough information about the current state of research in this field. The paper demonstrates an adequate understanding of the relevant literature in the field and cites an appropriate range of literature sources. All references cited are directly relevant to the research topic. In the Conclusions section, the results are presented clearly and analyzed appropriately. The authors have also provided directions for future research.

There are some minor remarks:

1) Please place Table 1, Table 4, and Figure 6 after they are mentioned in the text of the paper.

2) In the caption under Figure 3, please write "(b) Signal after band-pass filtering" instead of "(b) Band-pass filtering".

3) In the caption under Figure 5, please write "Block-diagram for grasping force control of prosthetic hand with the human in the control loop" instead of "Grasping force control of prosthetic hand with the human in the control loop".

The article is very good and relevant. I suggest that this article can be accepted in its current form.

Author Response

The authors would like to express their sincere appreciation to the reviewer for his/her constructive comments and suggestions, and his/her time and efforts spent in helping us to improve the quality and presentation of the paper.

  • Comment:Please place Table 1, Table 4, and Figure 6 after they are mentioned in the text of the paper.

Response: Thank you for your helpful comment and kind suggestion. We appreciate your suggestion regarding the placement of Table 1, Table 4, and Figure 6 in the manuscript. In accordance with your recommendation, we relocated Table 1, Table 4, and Figure 6 to be positioned after they are initially referenced in the text. This adjustment will ensure better coherence between the content and the corresponding visual aids, enhancing the overall readability and comprehension of the paper.

  • Comment:In the caption under Figure 3, please write "(b) Signal after band-pass filtering" instead of "(b) Band-pass filtering".

Response: Thanks for your kind comments. The caption under Figure 3 has been revised as suggested.

                   (a) Original EMG signal

          (b) Bandpass filtering

Figure 3. The myoelectric signal

  • Comment:In the caption under Figure 5, please write "Block-diagram for grasping force control of prosthetic hand with the human in the control loop" instead of "Grasping force control of prosthetic hand with the human in the control loop"

Response:Thanks for your heplful comment. We have modified Fig. 5 properly as follows:

Figure 5. Block-diagram for grasping force control of prosthetic hand with the human in the control loop

Reviewer 3 Report (New Reviewer)

Comments and Suggestions for Authors

This work presents a control framework, inspired by the human ventral-dorsal stream visual process, and primarily addresses functional grasping actions.

In my opinion, in this work there is not a scientific contribution to the modern state of the art, nor a feasible solution.

The experiments do not reproduce a real-life situation, and the system cannot be used for daily activities.

Author Response

The authors would like to express their sincere appreciation to the reviewer for his/her constructive comments and suggestions, and his/her time and efforts spent in helping us to improve the quality and presentation of the paper.

  • Comment: “This work presents a control framework, inspired by the human ventral-dorsal stream visual process, and primarily addresses functional grasping actions.

Response: Thank you for your kind comment.

  • Comment:In my opinion, in this work there is not a scientific contribution to the modern state of the art, nor a feasible solution.

Response: Thanks for kind comments. In light of this feedback, we thoroughly revisited our manuscript to better emphasize the scientific contributions and the feasibility of the proposed solution. We enhanced the clarity of our methodology, highlighted the innovative aspects, and provided a more comprehensive discussion of the practical implications of our work.

  • Comment:The experiments do not reproduce a real-life situation, and the system cannot be used for daily activities

Response: Thanks for your valuable comment. It is acknowledged that the current experiments represent an initial phase of validation and are not fully reflective of real-life scenarios or daily activities. The primary goal of these preliminary experiments was to assess the fundamental functionality and feasibility of the proposed system in a controlled environment.

In future iterations of our research, we intend to expand the scope of our experiments to encompass a more diverse range of real-life situations and activities. This will involve comprehensive testing that better simulates the dynamic and varied conditions users might encounter in their daily lives. The constructive criticism provided by the reviewer serves as valuable guidance, and we are committed to addressing these limitations in subsequent phases of our research.

This manuscript is a resubmission of an earlier submission. The following is a list of the peer review reports and author responses from that submission.

Round 1

Reviewer 1 Report

Comments and Suggestions for Authors

The paper presents a control strategy for prosthetic hands that aims at reducing the cognitive burden of the user.

The idea is to integrate an EMG-based prosthesis with a vision system to recognize the object to grasp and to call the appropriate motion primitive. Vocal commands are used in various stages of the process. Moreover, the system can give a real-time feedback about the exerted force.

There are two kinds of problems with the present paper, both about the design and the technical presentation.

Design problems:

1. The novelty of the proposed system is only in proposing a controller somehow inspired by the pathway of the human ventral-dorsal nerves. In fact, the introduction of vision systems to semi-automate the prosthesis control is around since more than 10 years.  The needed algorithms are not new. What is not yet ready for the real use is the equipment needed. Which camera is ready for this use? Where and how to position it (to wear a cask all the life is unfeasible)? And more important, will the user accept to rely on this more technological system?

2. The use of voice commands increases the hardware/software cost of the system and may create another kind of refuse, because the amputee has to control not his/her body but some external machinery. Moreover, it is unacceptable when the user is in a social situation.

3.The idea of force feedback through vision is not in line with reducing the cognitive burden.

4.The proposed experiments are far from convincing about the proposed solution. The experiments are only very preliminary, and are only qualitatively described. No quantitative results are reported, and only 2 cases have been done. The experimental setting is quite confused. A better design of the experiments and of the hypotheses to test is needed.

Technical description:

1. The technical description of the proposed method is unclear and largely imprecise. It names a series of hardware and software systems without any reference and without giving the details necessary to reproduce the results. Even the use of EMG signal is not described. How the signal is taken, with how many electrodes, etc.?

2. About the motion primitives, Table 2 and the following description are not in full agreement.

3. The sentence “It controls five servo motors respectively to realize the movement of the finger joint of the prosthetic hand.” is obviously referring to finger joints, not a single joint. However, it is not described the mechanical hand.

4. The meaning of the blobs in figure 4 that the user receives as feedback is not described. It seems to report the values of two sensors for each finger, while there is only one per finger.

In conclusion, about the fulfillment of the two main novelties (as indicated in the Introduction section):

- A controller inspired from the pathway of the human ventral- dorsal nerves. The process implemented does not introduce any new control strategy with respect to the traditional robotic approaches. The adoption of bioinspired methods of the visual cortex has already produced well known result both in vision and hand-eye coordination, not used here.

- The semi-autonomous controller, which gives some feedback and uses the EMG signals (human in the loop control), is not enough described to be reproduced.

Comments on the Quality of English Language

Finally, large editing is necessary.

The technical language is not properly used in some cases.

For instance at Line 171: analyze the object character … is characteristics

Reviewer 2 Report

Comments and Suggestions for Authors

The idea poposed in this manuscript is attractive and interesting for readers. However, I have a few concerns as folloes:

1- The manuscript is so long and wordy, without any mathematical modeling. This drawback should be justified by the respected authors.

2- The flowchart of Fig. 5 is very useful. However, its discussion should be provided.

3- Figure 6 should be discussed properly.

4- The authors should clearly dissuss the material which was used from the references and their new contribution to the model.

5- I think that adding some remarks on "limitation of the study" would provide a very sober review about what this study can do and waht cannot do.

Comments on the Quality of English Language

It is fine.